# Hot Workability of the Multi-Size SiC Particle-Reinforced 6013 Aluminum Matrix Composites

**DOI:** 10.3390/ma16020796

**Published:** 2023-01-13

**Authors:** Changlong Wu, Shuang Chen, Jie Tang, Dingfa Fu, Jie Teng, Fulin Jiang

**Affiliations:** 1College of Materials Science and Engineering, Hunan University, Changsha 410082, China; 2Hunan Provincial Key Laboratory of Vehicle Power and Transmission System, Hunan Institute of Engineering, Xiangtan 411104, China

**Keywords:** aluminum matrix composite, hot workability, processing map, microstructure, SiC particle, multidimensional composites

## Abstract

The size and distribution of ceramic particles in aluminum matrix composites have been reported to remarkably influence their properties. For a single ceramic particle, the particle size is too small and prone to agglomeration, which makes the mechanical properties of the composites worse. When the ceramic particle size is too large, the particles and alloy at the interface are not firmly bonded, and the effect of dispersion distribution is not achieved, which will also reduce the mechanical properties of the composites. The multi-size ceramic particles are expected to improve this situation, while their effect on hot workability is less studied. In this study, the hot deformation behavior, constitutive model, processing map and SEM microstructure were investigated to evaluate the hot workability of multi-size SiC particle-reinforced 6013 aluminum matrix composites. The results showed that the increased deformation temperature and decreased strain rate could decrease flow stresses. The flow stress behaviors of the composites can be described by the sine-hyperbolic Arrhenius equation with the deformation activation energy of *Q* = 205.863 kJ/mol. The constitutive equation of the composites is $\dot{\varepsilon \;}=3.11592\times 10^{13}{\left(\sinh\left(0.024909\sigma \right)\right)}^{4.12413}\exp\left(-\frac{205863}{RT}\right)$. Then, the hot processing map of the SiCp/6013 composites was constructed and verified by SEM observations. The rheological instability zone was in the region of a high strain rate. The optimal processing zone for composites was 450~500 °C and 0.03~0.25 s^−1^. In addition, the strain level was found to increase both the *Q* value and the area of the instability zone.

## 1. Introduction

Aluminum matrix composites are widely used in aerospace and automotive industries because of their high specific strength, high specific stiffness, and high wear resistance [1,2]. For the reinforcing phase of aluminum matrix composites, ceramic particles, such as SiC, Al_2_O_3_, TiB_2_, are often chosen because of their high hardness and high thermal conductivity [3,4,5]. However, due to the addition of such ceramic particles, the strength and hardness of aluminum matrix composites increase, while the plasticity and toughness generally decrease. It is reported in [6,7,8,9,10,11,12,13,14] that the size and content of ceramic particles had a remarkable influence on the properties that had to be adjusted to ensure uniform distribution of the reinforcement in the matrix. For instance, Wang et al. [7] examined the combined effects of particle size and distribution on the mechanical behavior of the SiC-reinforced Al–Cu alloy composites and found that by reducing the matrix/reinforcer particle size ratio, a more uniform distribution of SiC particles in the matrix is achieved, which can improve the properties. According to the review of Hashiguchi et al. [6], increasing SiC particulate loading in aluminum matrix composites increases the modulus rapidly. The further challenges focused on finer particles (<5 μm) and shortening of the mean free path between the ceramic particles and the aluminum matrix, which could result in remarkably improved mechanical strength. Du et al. [10] investigated the effect of the addition of RE and different sizes and amounts of SiCp on the activation energy, processing map, and strengthening mechanism and found that the size of the particles and the distance between the particles impacted the thermal stability of the composites. Gao et al. [5] investigated (TiC-TiB_2_)/Al-Cu-Mg-Si composites and found that multi-scaled sizes of TiB_2_ and TiC particles can effectively improve the tensile properties and wear resistance of the composites. Shen et al. [15] studied bimodal-size SiCp-reinforced AZ31B magnesium matrix composites and found that the addition of bimodal-scale SiCp can significantly refine the grains of the substrate because micron-scale SiCp promotes recrystallization nucleation, while smaller nanoscale SiCp impedes grain boundary motion allowing grain refinement.

Powder metallurgy is the most used process to prepare particle-reinforced aluminum matrix composite ingots. Then, hot working processes, such as hot extrusion and rolling, were generally employed to refine microstructure, reduce defects, and enhance properties. In addition, the thermal expansion coefficients of the matrix and reinforcing phases are different, thus making the hot processing properties of the composites poor. Therefore, it is necessary to study the hot processing properties of ceramic particle-reinforced aluminum matrix composites. In recent years, most studies concerned single-particle and hybrid-particle-reinforced aluminum matrix composites, where the reinforced particles were mostly in the range of tens of micrometers [16,17,18,19,20,21,22,23]. For instance, Shao et al. [24] studied constitutive flow behavior and hot workability of powder metallurgy processed 20 vol.%SiCp/2024Al composites. They found that the composites exhibited large regions of instability in the form of flow localization and cavitations located at the matrix/SiCp interfaces and within the SiCp clusters. Zhang et al. [25] fabricated the (micron+nano) bimodal-sized SiCp/Al2014 composites and found that the distribution and interfacial bonding of bimodal-sized SiCp to the aluminum matrix is superior to the single-size SiCp, resulting in superior mechanical properties. Chen et al. [17] studied the hot deformation behavior of hybrid aluminum matrix composites reinforced with micro-SiCp and nano TiB_2_. They found that the hyperbolic Arrhenius equation could describe the flow stress behaviors of the composites very well. The softening mechanism during the hot deformation of composites is dynamic recovery and dynamic recrystallization, and TiB_2_ nanoparticles promote the nucleation of dynamic recrystallization. Chen et al. [26] further studied the hot workability characteristics of Al–Si/SiCp + TiB_2_ hybrid aluminum matrix composites with various TiB_2_ contents. They found that the areas of the instability zone in the hot processing map of the hybrid composites expanded with the increase in the TiB_2_ content.

However, to date, studies on aluminum matrix composites with multiple sizes of single particles (especially for the ceramic particles less than about 10 μm) are rarely reported. Therefore, the study of (micron + submicron) SiC particle-reinforced aluminum matrix composites is still vast, especially with regard to its hot workability and the reasons for the instability of the composites. Therefore, in this study, the 20 wt.% SiCp/6013 aluminum matrix composites with three average sizes (0.7 + 2 + 5 μm) of SiC were prepared using powder metallurgy and hot extrusion. Then, hot compression experiments were employed to obtain the true stress–strain curves of these composites. Finally, the constitutive equations, hot processing map and microstructural observations were carried out to analyze the hot workability and deformation mechanisms.

## 2. Materials and Methods

Generally speaking, the size and content of ceramic particles in composites can be easily regulated by powder metallurgy. The powder metallurgy method can ensure uniform distribution of the reinforcement in the matrix, reduce agglomeration and segregation, and make the composites stronger. The disadvantage is that the dense material is not high. Therefore, the mechanical properties must be improved by extrusion, rolling and other processes. According to the literature and our previous work [11,27,28], if the content of ceramic particles is too small, the strengthening effect of ceramic particles will not be obvious, and the mechanical properties of composites will also be reduced if the weight fraction of the ceramic particles exceeds a certain level. Therefore, we determined the content of SiC in the composites as 20 wt%. The studied 20 wt.% SiCp/6013 aluminum matrix composites were prepared by powder metallurgical methods. Firstly, the 6013 aluminum alloy powder and three SiC powders of different sizes were weighed first and then put in the ball mill tank and mixed. The average size of 6013 aluminum alloy powder is 7 μm, and the average sizes of three kinds of SiC powder are approximately 0.7, 2 and 5 μm (with an equivalent ratio of 1:1:1). The purity rate and the dimensions of the powder particles are listed in Table 1. The proportion of 6013 aluminum alloy powder is 80 wt.%, and the proportion of SiC powder is 20 wt.%. Next, the mixed powder was cold pressed to the initial ingot, which was then sintered by vacuum hot pressing (80 MPa) for 6 h. Finally, the cylindrical composite ingot with a diameter of 85 mm was hot extruded to a sheet 50 mm wide and 10 mm thick at 480 °C. The manufacturing process of composites is shown in Figure 1a.

The cylindrical specimen with Φ 8 × 12 mm was machined from the extruded sheet in the extrusion direction for subsequent hot compression testing. The experimental schematic diagram is shown in Figure 1b. Cylindrical specimens are placed between two indenters equipped with mechanical sensors, and graphite sheet lubricant is placed between the specimens and the indenters to reduce friction. It can reduce the effect of uneven deformation caused by friction in the hot compression process. After the compression experiment starts, the Gleeble-3500 test system will automatically collect stress, strain, displacement and temperature data, and the collected true stress–strain data can be plotted by origin software to obtain the stress–strain curve. The specific parameters of hot compression are shown in Figure 1c, and the temperature was increased at a rate of 10 °C/s to the target temperature and held for 180 s to make the internal temperature of the specimen uniform. Then, the sample was compressed to a true strain of 0.7 at a given strain rate, and water was quenched immediately after the deformation to maintain the deformed microstructure. In addition, the microstructure of the composites after deformation was observed by scanning electron microscope (SEM) (FEI QUANTA 200). The particle size distribution of SiC powders was analyzed using a BT-9300S laser particle size distribution system.

## 3. Results and Discussion

### 3.1. Initial Microstructure of the Composites

As shown in Figure 2a–c, the SEM images of three different sizes of SiC particle powders correspond to 0.7, 2, and 5 μm, respectively. The microstructure of the multi-size SiC particle-reinforced 6013 aluminum composites prepared by powder metallurgy and hot extrusion is shown in Figure 2d–e. Based on SEM analysis, SiC particles are evenly distributed in the aluminum matrix: little aggregation phenomenon, good interfacial bonding between SiC particles and the aluminum matrix, and few void defects are indicated. From Figure 2d, it can be observed that the longer SiC particles have a characteristic distribution towards the extrusion direction. This may be due to the rotation of SiC particles along the extrusion direction during the extrusion process. From Figure 2e, it can clearly be seen that the SiC particles with different sizes are relatively uniformly distributed. The initial powder size distribution of three different SiC particles is shown in Figure 2f.

### 3.2. Flow Stress Characteristics

Figure 3 shows the true stress–strain curve of SiCp/6013 composites obtained from the hot compression experiment. At the beginning of deformation, the flow stress increases sharply with increasing strain and then reaches stability. At the same temperature, the greater the strain rate, the greater the stability stress achieved. At the same strain rate, the higher the deformation temperature, the lower the stability stress achieved. The results of the stress variation curve can be explained by the mechanism of process hardening and dynamic softening. At the beginning of processing, work hardening dominates. As the deformation continues, many dislocations are generated inside the composite material, cross-cutting and entanglement occur between different dislocations, and dislocation movement is blocked, making the stress increase significantly. In the second stage, the softening effect of dynamic reversion or dynamic recrystallization is enhanced, the stress growth slows down, and finally, the process hardening and dynamic softening effects reach a dynamic equilibrium, and the stress reaches the steady-state stage. Comparing the strain curves for different strain rates, it is found that the curves are wavy at lower strain rates, which is the result of competing process hardening and dynamic softening. The stress profile at a temperature of 350 °C with a strain rate of 5 s^−1^ tends to decrease slightly in the dynamic equilibrium state, which may be related to rheological instability, as discussed in the subsequent sections.

### 3.3. Constitutive Model

In the process of hot metallic deformation, in order to describe the high-temperature rheological stress σ at strain or steady state as a function of deformation temperature *T* and strain rate $\dot{\varepsilon }$, the Arrhenius instanton equation in hyperbolic sinusoidal form proposed by Sellars and Tegart [29,30] can usually be used. The equation usually has three forms:(1)$$\dot{\varepsilon }=A_{1}\sigma^{n_{1}}\exp\left(-\frac{Q}{RT}\right),\alpha \sigma <0.8$$
(2)$$\dot{\varepsilon }=A_{2}\exp\left(\beta \sigma \right)\exp\left(-\frac{Q}{RT}\right),\alpha \sigma >1.2\;$$
(3)$$\;\dot{\varepsilon }=A{\left(\sinh\left(\alpha \sigma \right)\right)}^{n}\exp\left(-\frac{Q}{RT}\right),for\;all\;\sigma \;$$

In general, the principal equations for low and high stresses can be described by Equations (1) and (2), respectively. Equation (3) applies to all stress states, where $\dot{\varepsilon }$ is the strain rate in s^−1^; *A*_1_, *A*_2_, *A* are structure factors; *σ* is the flow stress (MPa); *n*_1_ and *n* are the stress index; α is the stress level parameter, which is independent of deformation temperature, and α = β/n_1_. *Q* is the hot deformation activation energy (kJ/mol); *R* is the molar gas constant (8.31 J/mol K); *T* is the absolute temperature (K). The effect of deformation temperature and deformation strain rate on stress can be combined and expressed by the Zener–Hollomon (Z) parameter equation [31]. Equation (3) can be transformed into a hyperbolic sine function from Equation (4):(4)$$Z=\dot{\varepsilon }\exp\left(-\frac{Q}{RT}\right)=A{\left(\sinh\left(\alpha \sigma \right)\right)}^{n}$$

In order to calculate each parameter of the instanton equation, the natural logarithm is taken at both sides of Equations (1)–(4), and then the following equations can be obtained:(5)$$\ln(\dot{\varepsilon })=\ln\left(A_{1}\right)+n_{1}ln\sigma -\frac{Q}{RT}\;\;$$
(6)$$\ln(\dot{\varepsilon })=\ln\left(A_{2}\right)+\beta \sigma -\frac{Q}{RT}\;\;\;$$
(7)$$\ln(\dot{\varepsilon })=\ln A+nln\left[\sinh\left(\alpha \sigma \right)\right]-\frac{Q}{RT}\;\;\;\;$$
(8)$$\mathrm{lnZ}=\ln A+nln\left[\sinh\left(\alpha \sigma \right)\right]\;$$

The peak stresses of the flow stresses are used to calculate the parameters of the principal structure equation, substituting the peak stress of the SiCp/6013 composites and its corresponding different temperatures and different strain rates into Equations (5) and (6). The linear relationships of ln σ − $\ln(\dot{\varepsilon })$ and σ − $\ln(\dot{\varepsilon })$ at different temperatures are fitted, as shown in Figure 4. *n*_1_ and β are the slopes of the (a) and (b) curves in Figure 4, respectively. The mean values of *n*_1_ and β are thus calculated to be 7.03227 and 0.12448, respectively. The value of α can be calculated by the formula α = *n*/β, which is 0.017701254.

The conversion by Equation (7) yields:(9)$$Q=R{\left.\frac{\partial \ln\left[\sin h\left(\alpha \sigma \right)\right]}{\partial \left(1/T\right)}\right|}_{\dot{\varepsilon }}{\left.\frac{\partial \dot{\ln\varepsilon }}{\partial \ln\left[\sin h\left(\alpha \sigma \right)\right]}\right|}_{T}=Rnk\;\;\;$$

When *Q* is independent of temperature, the relationship between ln [sinh(ασ)] and 1/*T* is linear. As shown in Figure 4c,d, the values for *n*_2_ and *k*_1_ can be derived from the slopes of the ln [sinh(ασ)] − ln$\dot{\varepsilon }$ and 1000/T − ln [sinh(ασ)] plots, respectively. The mean values for *n*_2_ and *k*_1_ are calculated as 4.9974825 and 6.0068325, respectively. According to α = *n*/β, the new values can be obtained: *α*_2_ = 0.024908541 and *n*_3_ = 4.12413. Thus, *Q*_1_ = *Rn*_3_*k*_1_ = 205.863282 kJ/mol. Then, the above steps are repeated. By iterating the calculation several times, it yields *Q*_2_ = *Rn*_4_*k*_2_ = 205.7981154 kJ/mol, *Q*_3_ = *Rn*_5_*k*_3_ = 206.0114092 kJ/mol. It can be seen that the *Q* values calculated by the iterations do not differ much, so the *Q* values of the later constitutive equations are taken as *Q*_1_.

As shown in Figure 4e, lnA in Equation (8) can be obtained from the intercept (31.07013) of the lnZ -$ln\left[\sinh\left(\alpha \sigma \right)\right]$. Therefore, the value of *A* could be acquired as 3.11592 × 10^13^. The value of *R*^2^ for the fitted line is 0.99007, as shown in Figure 4e, which suggests that the hyperbolic sine model between the Z parameters is appropriate to describe the hot deformation behavior of the SiC_p_/6013 composites. Substituting the values of material parameters such as α, *n*, *Q* and *A* into Equation (3) yields:(10)$$\dot{\varepsilon }=3.11592\times 10^{13}{\left(\sinh\left(0.024909\sigma \right)\right)}^{4.12413}\exp\left(-\frac{205863}{RT}\right)\;\;$$

In order to verify the accuracy of the constitutive equation for the SiC_p_/6013 composites, the corresponding deformation temperature and strain rate are substituted into Equation (10). The calculated peak stress values from the intrinsic equation model of the SiC_p_/6013 composites are then obtained. The calculated and actual measured values are shown in Figure 4f. The error between the value calculated by the equation and the actual measured value is very small, so the constitutive equation can generally predict the high-temperature rheological stress of the SiC_p_/6013 composites well.

Considering that the strain also influences the parameters of the present constitutive equations, the parameters at different strain conditions are also calculated, and the results are summarized in Table 2. The hot deformation activation energy *Q* qualitatively reflects the energy potential of dislocations moving during the hot deformation process. The higher the *Q* value, the greater the driving force required to overcome the deformation resistance during hot deformation. From Table 1, the *Q* value increases with increasing strain, which indicates that the greater the strain of the SiC_p_/6013 composites, the greater the hot deformation activation energy required and the more difficult the deformation. The change in *Q* value is also related to the degree of recrystallization for dynamic softening. Then, the longer the time, the higher the degree of recrystallization, which leads to more grain boundaries and stronger obstruction to dislocation movement [32].

### 3.4. Processing Map

The processing map is based on a dynamic material model (DMM) and is superimposed by a power dissipation map and a rheological instability map [33,34]. The processing map and microstructure analysis are combined to further confirm whether the selected hot processing parameter conditions are reasonable. In DMM, the output energy *P* is composed of the dissipative quantity (*G*) and the dissipative co-efficient (*J*), as shown below:(11)$$P=\sigma \dot{\varepsilon }=G+J=\int_{0}^{\dot{\varepsilon }}\sigma d\dot{\varepsilon }+\int_{0}^{\sigma }\dot{\varepsilon }d\sigma \;\;\;\;\;\;$$
where *G* represents the energy consumed during hot deformation, most of which is dissipated as thermal energy and a small portion is stored inside the crystal in the form of crystal defect energy *J*, which represents the energy consumed by the tissue evolution during the deformation of the material, such as dynamic recovery, dynamic recrystallization and phase transformation. A power dissipation map can reveal the variation of the power dissipation factor (*η*) at different deformation temperatures and strain rates. *η* is defined as below:(12)$$\eta =\frac{J}{J_{max}}=\frac{2m}{m+1}\;$$
where *m* is the flow strain rate sensitivity index of flow stress. It is a function of temperature and strain rate. Based on the non-linear correlation between the value of strain rate sensitivity index *m* and strain rate $\dot{\varepsilon }$, a plastic stability judgment criterion is derived. Then the instability parameter *ξ* under various conditions could be defined as below:(13)$$\xi (\dot{\varepsilon })=\frac{\partial ln\left[m/\left(m+1\right)\right]}{\partial ln\dot{\varepsilon }}+m\;<\;0\;$$

Using Equation (13), the flow instability zone, or instability diagram, can be plotted for different temperatures and strain rates. The physical meaning of the above destabilization criterion is that if the system is unable to generate entropy at a rate greater than or equal to the applied velocity, it will cause the system to flow locally, resulting in flow instability [35]. The hot processing map could be obtained by superimposing the power dissipation and instability diagrams. The hot processing map can reveal the thermal processing performance of the material by identifying the steady-state rheological region, the rheological destabilization region, and the power dissipation coefficient under each processing condition.

Figure 5a shows the power dissipation diagram of the SiCp/6013 composites at peak stress based on the above construction method. The different colors represent the different dissipation efficiency in each deformation region, i.e., the highest dissipation efficiency in the red region and the decreasing dissipation efficiency from the red region to the blue region. According to the dynamic material model theory, the power dissipation coefficient represents the energy consumed by the tissue transformation of the composites during the deformation process. The higher value represents more energy consumed by the tissue transformation of the composites during hot processing and better tissue performance. It is also observed that there are two red regions, which are distributed in the higher temperature region 480~500 °C and a lower temperature region 350~360 °C, and the strain rate is between 0.05 and 0.5 s^−1^. Figure 5b shows the corresponding instability diagram. There are three regions with dissipation coefficients *ξ* < 0, which are three instability regions. Two destabilization zones are distributed at a high strain rate of 5 s^−1^ with temperatures of 500 and 350 °C, respectively. The area of these two instability zones is very small. The other region is distributed at temperatures between 400 and 420 °C and strain rates between 0.005 and 0.01 s^−1^. Figure 5c shows that the hot processing map with power dissipation and instability maps are superimposed. The gray area in the period is the instability zone. According to this processing map, the distribution of the optimal processing parameters of the SiCp/6013 composites are the two red areas in the processing map where no rheological instability occurs.

Figure 6 shows the hot processing map of the SiCp/6013 composites under different strain conditions. Figure 6a shows the hot processing map with a strain of 0.1. There is only a smaller area in the rheological instability zone, called instability zone 1, where the temperature is between 380 and 410 °C, and the strain rate is between 0.005 and 0.01 s^−1^. As the strain increases, instability zone 1 disappears at a strain of 0.3, and two new instability zones appear, respectively, which are instability zone 2 and instability zone 3. Both instability zones are distributed in the high strain rate zone between 0.5 and 5 s^−1^, and the temperature is distributed between 350 and 370 °C and 420 and 500 °C, respectively. As the strain increases to 0.5, both instability zone 2 and instability zone 3 expand. When the strain increases to 0.7, the rheological instability zone expands further, and almost all the regions with a strain rate greater than 5 s^−1^ are destabilized. The real stress–strain curve of the composites also verifies this, and the curve with a strain of 5 s^−1^ has a stress drop when the strain almost reaches 0.7, indicating that rheological instability occurs at this time.

### 3.5. Microstructural Characterization

In order to verify the accuracy of the hot processing map, some deformed samples located in the destabilized and non-destabilized areas were selected for SEM observation. Figure 7 shows the microscopic SEM images of the samples with different deformation conditions at a strain of 0.7. Figure 7a–d shows the organization of hot compressed samples with strain rates of 5 s^−1^ at different temperatures, respectively. It can be seen that the SiC particles in the rheological destabilization zone are broken, and the SiC particles are broken severely at lower temperatures. The degree of SiC particle breakage decreases as the temperature increases. At 350 °C, almost all of the SiC particles are broken, and the broken SiC particles are agglomerated together. The accumulation of broken SiC particles leads the stress to drop when the strain is about 0.7. At 450 °C, it can be observed that only a small amount of breakage occurs in the larger SiC particles of the composites. At 500 °C, it can be observed that most of the SiC particles are relatively intact in appearance, but small holes and crack defects appear in the aluminum matrix. The reason for this is that the strain rate is too high, and the deformation of the SiC particles in the reinforced phase cannot keep up with the deformation of the matrix resulting in the fragmentation of SiC particles. The high strain rate may also affect the interfacial bonding between the SiC particles and the aluminum matrix, resulting in interfacial cracking and holes between the SiC particles and the aluminum matrix, which leads to rheological instability and a decrease in stress. The decrease in the degree of SiC particle fragmentation at higher temperatures is due to the fact that higher temperatures provide enough energy to activate hot deformation. The higher the temperature, the stronger the atomic diffusion ability, and thus the deformation ability rises. Therefore, the accuracy of the hot processing map is verified from both the stress–strain curve and the SEM microstructures of the SiCp/6013 composites. The rheological instability zones of the hot processing map should be avoided in the actual hot processing production. Furthermore, the region with a high power dissipation coefficient should be selected for hot processing, i.e., the red region of the hot processing map.

## 4. Conclusions

In this work, a multi-size SiC particle-reinforced 6013 aluminum matrix composite was prepared. Hot deformation behavior, a constitutive model, a processing map and SEM microstructures were studied to evaluate the hot workability of the composites. The conclusions can be summarized as follows:

1. The true stress of the SiCp/6013 composites increased rapidly to a peak value with increasing strain and then stabilized. The peak stress decreased with increasing temperature and increased with increasing strain rate.

2. The flow stresses of the SiCp/6013 composites could be expressed by the sine-hyperbolic Arrhenius equation, which was a good predictor of the rheological stress at stabilization. The constitutive equation of the SiCp/6013 composites is $\dot{\varepsilon }=3.11592\times 10^{13}{\left(\sinh\left(0.024909\sigma \right)\right)}^{4.12413}\exp\left(-\frac{205863}{RT}\right)$. In addition, the hot deformation activation energy of the composites under different strain levels was estimated, which increased slightly with increasing strain.

3. The hot processing map of the SiCp/6013 composites was constructed and verified by SEM observations. The rheological instability zone was in the region of a high strain rate. The area of the instability zone increased with the strain. The optimal processing zones for composites were 450~500 °C and 0.03~0.25 s^−1^. Deformation flow instability of composites at low temperatures is mainly characterized by fragmentation and fragment agglomeration of reinforcing particles. The instabilizing features at high temperatures are mainly holes and cracks. Fragmentation of SiC particles in the instability zone and aggregation of fragments lead to a stress drop.

## Figures and Tables

**Figure 1 materials-16-00796-f001:**
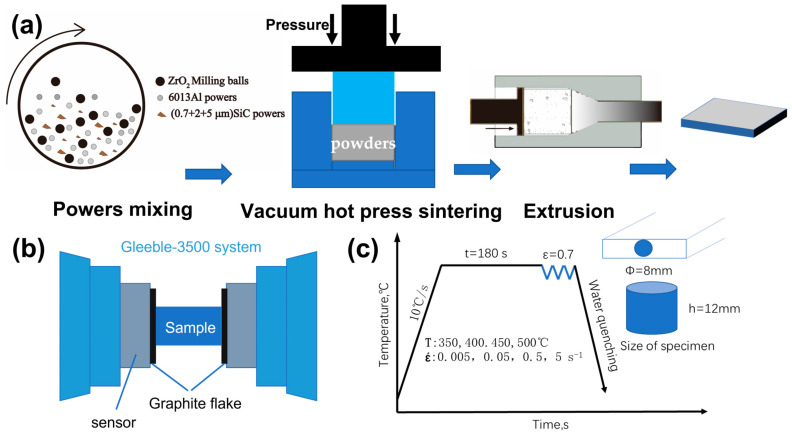
(**a**) Schematic diagram of the composite manufacturing process; (**b**) schematic diagram of hot compression test on Gleeble-3500 simulator; (**c**) hot compression parameters and the shape dimensions of the compressed specimen.

**Figure 2 materials-16-00796-f002:**
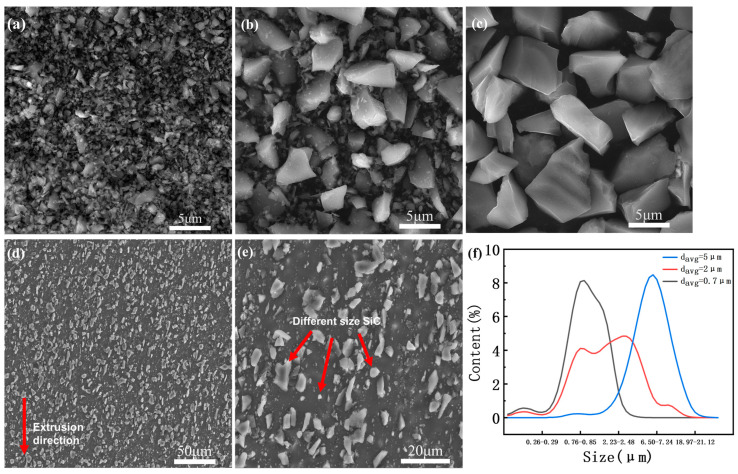
SEM images of initial microstructure:(**a**) 0.7 μm SiC powders, (**b**) 2 μm SiC powders, (**c**) 5 μm SiC powders, (**d**,**e**) as-extruded 20 wt.% SiCp/6013 aluminum matrix composites, (**f**) Particle size distribution of SiC powders.

**Figure 3 materials-16-00796-f003:**
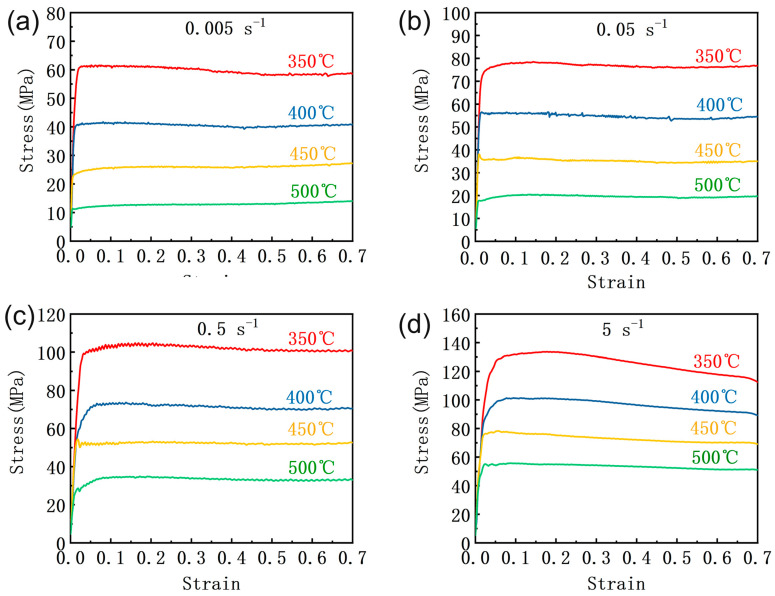
True stress–strain curves of the SiCp/6013 composites under different compression conditions: (**a**) 0.005 s^−1^; (**b**) 0.05 s^−1^; (**c**) 0.5 s^−1^; (**d**) 5 s^−1^.

**Figure 4 materials-16-00796-f004:**
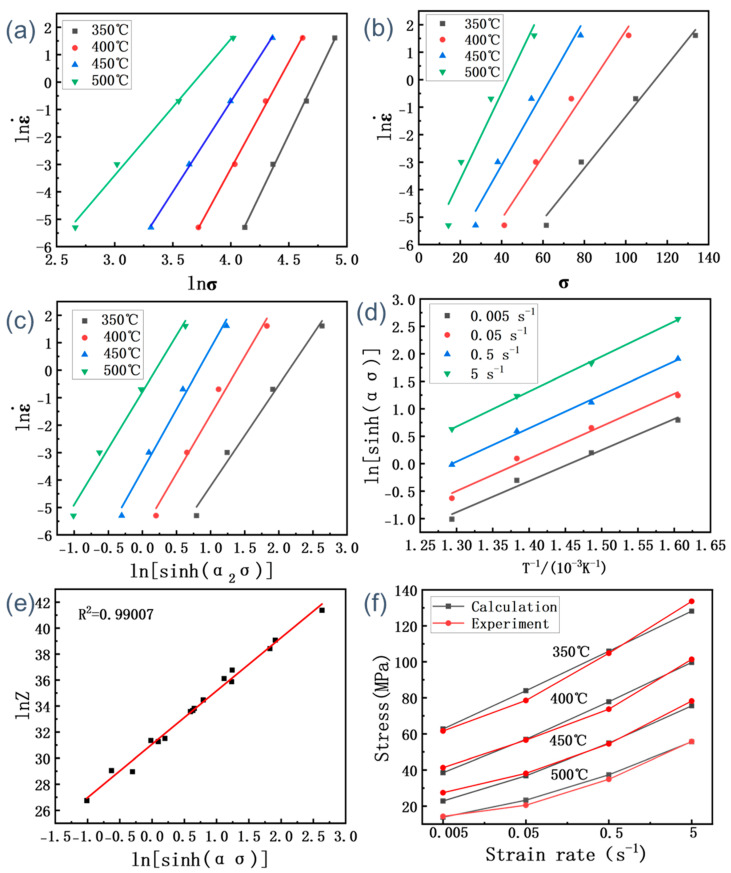
Relationships between strain rates and peak stress for the SiCp/6013 composites: (**a**) ln σ − ln$\dot{\varepsilon }$; (**b**) σ − ln$\dot{\varepsilon }$; (**c**) ln [sinh (ασ)] − ln$\dot{\varepsilon }$; (**d**) T^−1^/10^−3^K − ln [sinh (ασ)]; (**e**) ln [sinh (ασ)] − lnZ; (**f**) Comparison of calculated and experimental stresses.

**Figure 5 materials-16-00796-f005:**
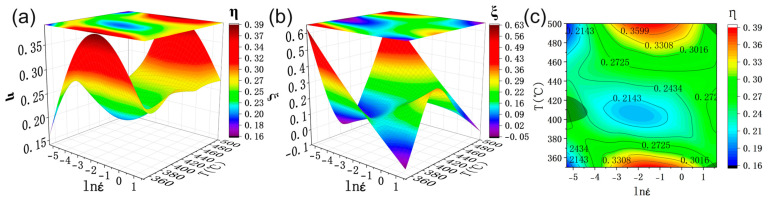
Processing diagram at peak stress: (**a**) power dissipation 3D map; (**b**) instability map; (**c**) processing map.

**Figure 6 materials-16-00796-f006:**
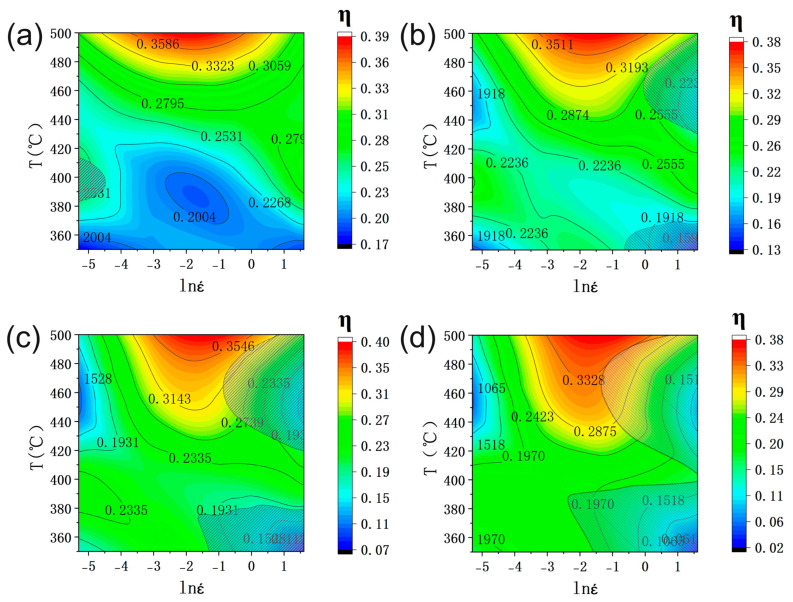
DMM processing maps of the SiCp/6013 composites at different strain levels: (**a**) ε = 0.1, (**b**) ε = 0.3, (**c**) ε = 0.5, (**d**) ε = 0.7.

**Figure 7 materials-16-00796-f007:**
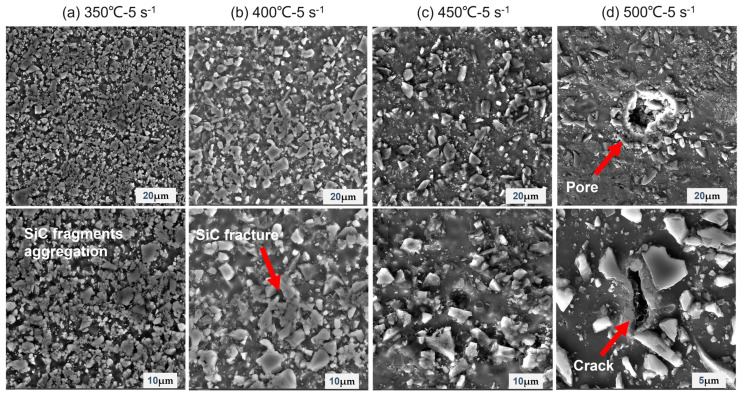
SEM images of the SiCp/6013 composites for instability zones at a strain of 0.7: (**a**) 350 °C-5 s^−1^; (**b**) 400 °C-5 s^−1^; (**c**) 450 °C-5 s^−1^; (**d**) 500 °C-5 s^−1^.

**Table 1 materials-16-00796-t001:** The purity rate and dimensions of powders.

Powder	Purity (%)	Dimension (μm)
6013Al	99.2	7
SiC	99.5	0.7
SiC	99.5	2
SiC	99.5	5

**Table 2 materials-16-00796-t002:** Parameters of the constitutive equation under different strains for the SiCp/6013 composites.

Strain ε	A (MPa^−1^)	*n*	*A* (s^−1^)	*Q* (kJ/mol)
0.1	0.035486657	3.15642	6.67 × 10^12^	205.1840524
0.3	0.036103	3.229925	1.69 × 10^13^	211.1135983
0.5	0.036829023	3.3890925	3.83 × 10^13^	216.6694489
0.7	0.036537749	3.672925	1.67 × 10^14^	226.9465338

## Data Availability

Not applicable.

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
