# Peer review of "Hot Workability of the Multi-Size SiC Particle-Reinforced 6013 Aluminum Matrix Composites"

_materials, 2023, doi:10.3390/ma16020796_

Round 1

Reviewer 1 Report

SUMMARY

The article submitted for review is devoted to a topical issue. The hot workability of multi-size SiC particle reinforced 6013 aluminum matrix composites is considered. The relevance lies in the fact that the distribution sizes of ceramic particles in composites with an aluminum matrix significantly affect the properties, while their effect on hot workability is poorly understood. The study presents interesting methods by which work was carried out to solve this scientific problem. The authors obtained interesting results, found that an increased deformation temperature, a reduced deformation rate can reduce the flow stress. All the necessary equations were presented, maps were drawn, rheological instability was considered. Thus, the authors have done a great job, eliminating a serious scientific deficit. The article has a certain scientific novelty and practical significance, while the article has shortcomings, they should be corrected. They are presented below.

COMMENTS

1.     The abstract presented by the authors does not reflect the scientific problem. The authors say that the influence of the size and distribution of ceramic particles on hot workability is poorly understood, but do not explain what specific scientific problem they are formulating. That is, are there any prerequisites for the fact that this poor knowledge leads to a large number of defects, failures and similar negative phenomena. That is, the authors must link the scientific and practical problem and bring this wording at the beginning of the abstract.

2.     The next observation is that the authors have described the test methods in great detail, but did not disclose in a specific way which research methods were applied. Authors should pay less attention to the details of the description of the test methods, and focus more on the general characterization of the methods and all iterations carried out at the research level.

3.     The result is presented poorly, only quantitatively. I would like to see a more fundamental formulation of the achieved scientific result.

4.     The keywords chosen are rather uninformative, the authors should think over those terms that will help scientists interested in the topic “Hot Workability of Multidimensional Composites with Aluminum Matrix and Silicon Carbide Particles” to find interesting information for themselves. Thus, the keywords need some adjustment.

5.     The authors performed a very poor review of the literature. The current state of the issue is poorly reflected. The authors should analyze more sources and reveal the depth of the scientific problem. In addition, a clear goal and tasks of the study are needed. In Section 2, the authors go straight to the description of the preparation of aluminum matrix composites, but they did not present a program of experimental studies. In general, the methodology is described only superficially, the materials are presented very uninformatively. Should be improved.

6.     The results in the "Results and discussions" section look somewhat uninformative. In the graph in Figure 4, it might be necessary to increase the number of points, or to interpret the obtained dependences more correctly, especially considering that many of them are approximated as straight lines. It is necessary to describe this amount of graphic material in more detail.

7.     As for Figure 3, the authors presented them in a rather low quality and the presented graphical results are poorly distinguishable on them.

8.     The SEM analysis in Figure 2 needs further interpretation. Figures 5 and 6 are presented in very low quality, many symbols in these figures are indistinguishable to the naked eye. The discussion of the results obtained should include a detailed comparison with the results of other authors. Also, the authors did not do enough work to discuss the results obtained.

9.     Conclusions need some refinement. A more specific description of the obtained scientific result is needed. It should also reflect the prospects for the development of the study and what new knowledge and existing ideas have been obtained and developed.

10.  The list of references from 22 sources is very small for such a topic, it should be increased to 30–35 sources.

11.  In general, the article needs some adjustment of style and editing of the English language.

Reviewer 2 Report

The authors presented an article about “Hot Workability Of the Multi-size SiC Particle Reinforced 6013 Aluminum Matrix Composites”. The authors have done a good study on hot workability for composite material. It can be said that the authors should be congratulated for this. I think the paper is well organized and is appropriate for “Materials” journal but the paper will be ready for publication after major revision.

·       The abstract looks good. Please include significance numerical results

·       For the introduction section, please add more reference (related to the topic of the article) and briefly explain them.

·       In the last paragraph of the introduction, it should be expressed the novelty of the study, the differences from the past in detail.

·       The red inscriptions in the figure (Ex: Fig. 1 and Fig. 7) and some figure inscriptions (Ex: Fig. 5) are not clear. Please make the figures better quality and clear.

·       How did the authors determine the 20 wt.% ratio for the composite material? Please mention in the material method section.

·       What are the purity rates of the powders? Please prepare a table containing the purity rate and the dimensions of the powder particles and present it in the article.

·       It is possible to obtain SEM images of each dust particle. This image can give information about the dimensions of the powders. In addition, supporting these images with EDS analysis will make the article more attractive.

·       How was the stress-strain curve obtained? Please provide more information in the material method section.

·       Present a schematic picture of an experiment set related to the experiments performed in the article study in the article.

·       Improve the conclusion parts.

·       Please fix the typographical and eventual language problems in paper.

·       The paper is well-organized yet there is a reference problem. First, your reference list contains no paper from “Materials” journal. If your work is convenient for this journal’s context then there are many references from this journal. Secondly, cited sources should be primary ones. Namely, indexed area shows the power of a paper and directly your paper’s reliability. Please make regulations in this direction.

*** Authors must consider them properly before submitting the revised manuscript. A point-by-point reply is required when the revised files are submitted.

Reviewer 3 Report

The authors investigated the hot workability of the multi-size SiC particle reinforced 6013 aluminum matrix composites. However, the goal of the study is not clear. I have some comments to improve the quality of the manuscript as below:

1. SEM images of SiC powders with different sizes should be provided.

2. The effect of the  SiC powder with different sizes on the hot workability of the composite should be investigated and presented.

3. The weight percent of the individual SiC powder (0.7, 2, 5 micron) in total 20 wt.% SiC of the composite should be also provided. How effect of them?

4. SEM images in Figure 7 should be presented at the same scale bar to compare.

5. Please explain why the authors have to mix different SiC powder together for the investigation. What different with the composites containing single SiC powder. 

Round 2

Reviewer 2 Report

Thank you for revision

Reviewer 3 Report

The manuscript could be accepted in its current form.